# Study of the Changing Relationship between Religion and the Digital Continent—In the Context of a COVID-19 Pandemic

**Jean Marc Barreau**

Succursale Centre-Ville, Faculté des Arts et des Sciences, Pavillon Marguerite-d'Youville,
Institut d'Études Religieuses, University of Montreal, Montréal, QC H3C 3J7, Canada;
jean.marc.barreau@umontreal.ca

**Abstract:** This article proposes to study the changing relationship between religion and the digital continent as a result of the COVID-19 pandemic. To achieve this objective, the paper is divided into three parts. First, it offers an overview of the connection between religion and the digital environment, outlining four possible paradigms of the open relationship between these two worlds. Second, the article discusses the research project undertaken during the COVID-19 pandemic on behalf of the Corporation of Thanatologists of Quebec, focusing on the relationship between delayed funerals and delayed grief. In particular, this article deals with one of the solutions proposed to thanatologists, i.e., the development of a culture of bimodal ritual, both in person and remote, and therefore partly digital. Using this solution as a pointer, religion's shift toward digital technology in the COVID-19 period is analyzed in the third part of the article. To this end, the four paradigms drawn from the overview are set against the research focus areas resulting from the solution proposed to the Corporation of Thanatologists.

**Keywords:** religions; digital continent; context of a COVID-19 pandemic





## 1. Introduction

The successive waves of the COVID-19 pandemic and the emergence of multiple virus variants have shaken up a number of institutions and ways of life, both societal and personal, with almost unprecedented demands for adaptation. The religious and funerary environments have not been spared. Funeral home directors had to respect unheard of draconian sanitary measures imposed by authorities and to implement them with the bereaved families who requested their services. It is in this context that in June 2020, the Corporation of Thanatologists of Quebec, which currently has more than 500 branches throughout the province, asked me to conduct a study on the relationship between delayed grief and delayed funeral. One of the solutions proposed to the corporation was the implementation of bimodal rituals, in person and remote, using a digital medium. In this article, this suggestion is used to anchor theoretical reflection in the pandemic context and to serve the general purpose of the study which is to explore the way in which the COVID-19 pandemic induces a shift in the relationship between the religious and digital realms and the consequences thereof.

The first step in doing this will be to provide an overview of the relationship between the religious and digital environments from which will be drawn four essential paradigms for analyzing the changing relationship between these two worlds. The second step will be to present the solution developed in prior research, i.e., bimodal ritual, as a pointer for examining the shifting relationship between the two spheres. Prospects for future research will be brought up in the conclusion.

## 2. Overview

For the purpose of this article, religion is defined from a religious science standpoint, following two authors in particular. According to Campbell, "Religion here refers to

organized systems of spiritual beliefs" (Campbell 2005, p. 5), whereas Douyère looks at religion from the perspective of "communication and transmission of information, dogmatic and ritualistic, suggesting the presence of the divine" (Douyère 2016, p. 3).

It is worth noting that the expression "digital continent" found in the title is widely used in academic and religious circles, in particular in the humanities and social sciences. For example, and by way of introduction, this is largely thanks to Pope Emeritus Benedict XVI and the universal nature of his message, but also because of the apparent anachronism between the institution he represented and the virtual reality to which he referred. More specifically, it was on the occasion of the 44th World Day of Social Communications, on 16 May 2010, that he delivered a universal message with the eloquent title "The Priest and Pastoral Ministry in a Digital World: New Media at the Service of the Word." In it, he acknowledged the existence of a "digital continent" (p. 3) and the importance of a "pastoral presence in the world of digital communications" (p. 3). Insisting on the responsibility to announce the Word of God (p. 3), the German theologian presented this "new" world as a space "with its almost limitless expressive capacity" (p. 2). His posture is clear: it is the priests' duty "to be present in the world of digital communications as faithful witnesses to the Gospel, exercising their proper role as leaders of communities which increasingly express themselves with the different 'voices' provided by the digital marketplace" (p. 2). The goal being to make God's love more concrete and rooted in the present (p. 2), to open the way for new encounters (p. 3). According to the theologian, "The development of new technologies and the larger digital world represents a great resource for humanity as a whole and for every individual, and it can act as a stimulus to encounter and dialogue." In similar circumstances, Pope Francis did not hesitate to call "the Internet, a gift from God" (Pope Francis 2014, p. 2).

This introduction to the relationship between the religious world and the digital continent makes room for Campbell's systematization into four distinct paradigms. Professor of communications in Texas and specialist of what is referred to as digital religion, Campbell is the author of a groundbreaking article: "Spiritualising the Internet. Uncovering Discourses and Narratives of Religious Internet Usage" (Campbell 2005). I will expand thereon, drawing on the analysis offered by Douyère, another specialist in the field, in his review article "De la mobilisation de la communication numérique par les religions" (Douyère 2015) (how religions harness digital communication) in order to complete the overview of the changing relationship between the world of religion and the digital continent.

*Four Paradigms*

Although the expression "digital continent" is used by Maignant (2017) in the title of an article he penned analyzing the tensions experienced within the Catholic Church in Ireland around the use of the digital continent, the author refers more specifically to Campbell's paper (2005). The latter is essential to our reflection because it offers a systematic analysis of the phenomenon of "Spiritualising the Internet" (Campbell 2005, p. 2) according to a qualitative method: "Four common discourses used by religious Internet users to conceive of and describe the Internet, along with four corresponding narratives of religious Internet use are introduced (p. 2). The four discourses develop "why" the digital Web can serve the religious sphere and the four narratives analyze "how" this is done (p. 14). The "domestication" approach is central, as it underscores the way religions gradually and familiarly appropriate digital culture by adapting it to their various needs (pp. 4–5), i.e., unawares and revealing in the process features of their own.

## 3. Four Discourses and Four Narratives

The first of the four discourses looks at the "Internet as a spiritual medium facilitating spiritual experience" (Campbell 2005, p. 10). This is about integrating the concept of "meaning significance" (p. 10) insofar as technology is an extension of spirituality and that religious engagement with the digital world will facilitate spiritual encounters (p. 10). In addition, the author cites Jennifer Cobb's (1998) book *Cybergrace* in which Cobb links

the digital world to Teilhard de Chardin's holistic theology: the digital space becomes a genuine spiritual network (Campbell 2005, p. 11). The second discourse considers the "sacramental space suitable for religious use" (p. 11). Unlike the first discourse which claims the Internet as a spiritual space, here the World Wide Web is not seen as a holy space "by nature" (p. 11), but as having the potential to become one if it is the place where "ritual" is performed (p. 11). Online rituals therefore can turn cyberspace into a spiritual place. As examples, the author cites "cyber-churches, cyber temples or virtual shrines" (p. 11), digital spaces where discourses affirm that the "Internet can create a sacred space so it can be used for religious purposes" and go all the way to forming virtual religious communities (p. 11). The author also refers to Wertheim's (2000) book *The Pearly Gates of Cyberspace*, where she claims cyberspace to be a non-physical space replacing gothic cathedrals (Campbell 2005, p. 11). The third discourse describes the Internet, as a "tool to promote religion and religious practice" (p. 12). Here, the tool, an otherwise "neutral artefact" is entirely dependent on the motives and desires of its users (p. 12). As such, it can serve three functions: seeking religious information, fostering spiritual relationships and reconfiguring traditional religious activities. Referring to Walter Wilson's book *The Internet Church* published in 2000, Campbell shows that cyberspace can be a new terrain for proselytizing endeavours where actors believe they receive the divine mandate to convert without any limitations, be they temporal or cultural (p. 12). The fourth and last discourse analyzed by Campbell perceives the Web as a "technology for affirming religious life" (p. 13). A discourse that insists on the way in which the Internet can connect those of the same religious tradition across distance, time or other limiting factors (p. 13). A reality that enables the conception of a global community of followers (p. 13). Professor Campbell cites Benda Brasher who argues in her 2001 book *Give Me That Online Religion* that the Internet allows for "new forms of traditional religion expression, highlighting examples such as a cyber-seder or a virtual Passover as a way of helping people reconnect with their Jewish faith" (p. 13). Specifically, the fluidity and autonomy of the Internet, according to Brasher, allow to embrace a religious heritage of the past, to contribute to a global brotherhood, and finally, to facilitate an interreligious culture (p. 13).

For the sake of synthesis, it seems appropriate to represent these four discourses in a figure (Figure 1) that characterizes the relationship between religion and the digital continent from the perspective of its purpose, the "Why" explored by Campbell.

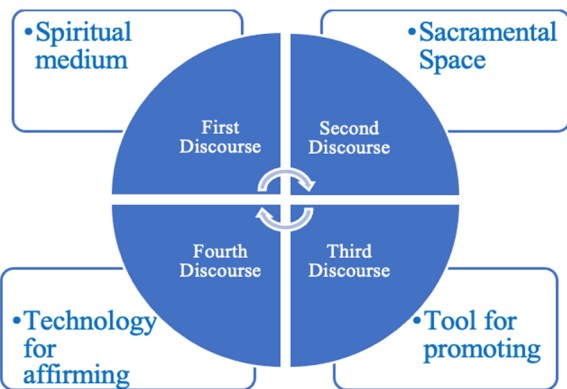

**Figure 1.** Four common discourses.

Whereas "the discourses" presented provide a framework for explaining why the Internet can be used in religious activities, Campbell opines that they do not fully capture how the Internet is shaped and employed by religious users (p. 14). As a result, connecting "discourse" and "narrative," the "why" and the "how," helps to clarify the extent to which the digital environment offers unique opportunities for religious settings. This relationship is systematized in Figure 2 Discourses and narratives.

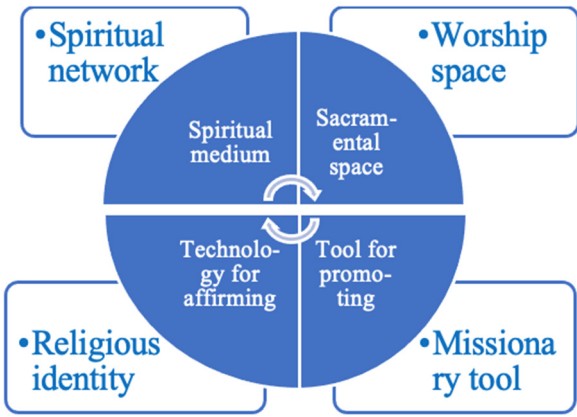

**Figure 2.** Discourses and narratives.

From this initial treatment of the question, we derive an understanding of the way in which religion is ontologically positioned in relation to the digital continent. In fact, the first paradigm–Spiritual network–situates religion "outside" the digital continent, considering the latter as an extension of the former: Spiritual medium (Table 1). The second paradigm–Worship space–places religion "inside" the digital continent: Sacramental space. The third paradigm–Missionary tool–places religion downstream from the digital continent: Tool for promoting. The fourth paradigm–Religious identity–places religion upstream from the digital continent: Technology for affirming. These four paradigms and postures presented below in the Table 1 will underpin my analysis.

**Table 1.** Paradigms and postures.

| Paradigms | Postures |
| --- | --- |
| Spiritual network | Exterior posture |
| Worship space | Inner posture |
| Missionary tool | Posture after |
| Religious identity | Front Posture |

*Digital Performativity*

Douyère (2015) builds his conception of religion around the "communication and transmission of information, dogmatic and ritualistic, suggesting the presence of the divine" (p. 3). The epistemological basis adopted here is one of performativity. Consequently, the entire set of means of communication falls prey to religions and provides a "space of signification" and unique effectiveness. In his view, religion maintains a grip on the digital environment through "text, rite, image discourse or absence thereof" (p. 3). Digital Information and Communication Technologies (DICT) therefore offer a particularly effective modern info-communicational assistance to religious practice (p. 3). To introduce his subject, Douyère underscores the way in which DICT enable renewed religious "apologetics" and "propaganda" as tools for a new form of proselytism allowing to reach out to new populations while offering to the masses faith content until then reserved to a handful of scholars (p. 3). I will draw on his mapping of religions that use DICT in order to synthetize the manner in which various religious groups employ them and finally adopt the nine-point typology of the uses that religions make of DICT.

Among the sources that Douyère draws on for his mapping of religions integrating DICT are Campbell (2010) and Podselver (2015), both of whom studied Judaism's connection to DICT, and Niculescu (2015), who examines Jewish use of Buddhist mindfulness meditation via the Internet. Douyère also cites research conducted by Bunt (2009), Roy (2000), Wheeler (2014), Lövheim (2015) and Varlik (2015) on Shi'a law in relation to the Internet. Based on Douyère's literature review, Campbell emerges as an expert in the field. She is widely cited, including in a study of Turkish Islam in relation to the Internet, and

in studies on Tibetan Buddhism and Japanese religions. Douyère also highlights a study carried out by Duteil-Ogata (2015) on the creation of Japanese funerary rites thanks to computers and digital technology, and mentions Vekemans (2014) who focused her studies on digital Jainism (p. 5). References concerning Protestantism and Catholicism are abundant, including research conducted by Mottier (2015) on Internet use in evangelism and in worship in a charismatic Pentecostal church, by Vanel (2015) on the relationship between Mormonism and the Internet, and by Julliard (2015) on Internet use by the Anglican Church (p. 5), to mention a few.

Aside from mapping, Douyère proposes a literature review in order to grasp the way in which religion appropriates DICT. In the American literature (Helland, Campbell, Howard) there seems to be a consensus on the question of the "community formed or assisted by DICT" (p. 6). As for the French literature (Jonveaux, Bratosin, Tudoret Coman, Douyère), it focuses primarily on three areas: the institutional dimension of religious use of DICT; the innovative aspect of devices; and with a "quasi-philosophical" outlook, the dimension of "mobilized subjectivity" (p. 6).

More broadly, let's keep in mind the distinction made by Helland (2005) between "online religion" and "religion online" (p. 7), as well as the notion proposed by Howard (2011) in the aftermath of the events of 11 September 2001, of a virtual community "of interpretation through exchange" that gave rise to the concept of "newsgroups" or "imagined community" which has a rich "prophetic" dimension and is free from all forms of authority. This prophetic aspect transmitted by the digital continent is specified and enhanced through "pedagogical," "meditative," "mystagogical" and "semiotic" dimensions (p. 9). Furthermore, Douyère draws attention to Bratosin et al. (2010) who associate the practice of the sacred within the digital environment with the birth of a new common "norm" that is akin to the concept of the "undifferentiated believer" (p. 10). This leads the believer to identify with a digital space: "Ego-localization" (p. 11).

Finally, Douyère puts forward a typology of the uses that religious environments make of DICT, identifying nine categories: "prayer practices," "legalistic practices," "religious socialization practices," "knowledge practices," "visual and discursive practices," "practices relating to guiding religious action," "informational practices," "political practices" and "action-inducing practices" (p. 14). We present this typology below in Table 2.

**Table 2.** Typology of DICT usage.

| Category 1 | Prayer practices |
|---|---|
| Category 2 | Legalistic practices |
| Category 3 | Religious socialization |
| Category 4 | Knowledge practices |
| Category 5 | Visual and discursive practices |
| Category 6 | Religious action guidance |
| Category 7 | Informational practices |
| Category 8 | Political practices |
| Category 9 | Call to action |

The mapping of religious use of DICT, the review of American and French literature, and the typology converge as regards the performative nature of this use, sometimes going so far as to assume a proselytizing discourse or posture. Moreover, Douyère shows how DICT can in turn appropriate the religious world to the point of modifying it in part (p. 15). Religion's traditional territorial dimension is being opened to a geography without borders and its traditional temporal dimension is also totally transformed insofar as "knowledge" and "events" are widely and instantaneously available. Consequently, the territorial and temporal dimensions reverse the relationship to cyberspace. Here, it is DICT that modify the configuration of the religious sphere to the point of sometimes eliminating the mediations or intermediaries that mark traditional religion. Douyère speaks of a possible "disintermediation" (disappearance of intermediaries), at best of a "shift of religious intermediation" (p. 15). Whereas Campbell's four paradigms define the

relationship between religion and cyberspace, the concept of disintermediation put forward by Douyère analyzes the performativity of DICT, and more specifically the potential bypassing of intermediaries and even religious authority and the so-called traditional sacraments.

## 4. Research Project

The research project undertaken on behalf of the Corporation of Thanatologists of Quebec began in June 2020 and it will be carried out over a period of two consecutive academic years. The first part focused on the impact of delayed funerals on the grieving of bereaved families during the COVID-19 pandemic and on possible solutions. The path explored in the study involves the implementation of bimodal funeral rituals (therefore partly virtual) in order to alleviate delayed grief and it speaks to the general purpose of this article which is the changing relationship between religion and the digital continent as a result of COVID 19, insofar as the solution considered can serve as a marker of such a paradigm shift.

### 4.1. Presentation

The initial stage of this research focused on the study of the impact of delayed funerals on mourning and the so-called delayed grief of bereaved families who sought the services of the Corporation of Thanatologists between June 2020 and January 2021. A mixed methods research design (MMR) was used for this purpose, involving a quantitative model for one phase of the research and a qualitative model for the other (Anadón 2019, p. 106). Specifically, MMR is characterized by the quantitative/qualitative continuum and by the fact that as extremes are approached (in this case, bereaved families), the mixed methods design will have either a quantitative or qualitative dominance trend (p. 107). This dual character and flexibility of the research guided the drafting of the four initial questionnaires. Due to the ongoing pandemic, the survey method preferred was an online questionnaire administered through a professional website that respected the scientific content of the research (Gauthier and Bourgeois 2016, p. 477). The questionnaires were developed according to the four major steps required in social science. In particular, essential concepts, i.e., ritual and complicated grief, were identified on the basis of the research question (1), a funnel-shaped question sequence was established (3) and finally, a pre-test was carried out (4) with a group of university students (p. 483). In each of the four questionnaires, the alternation between semi-open and closed questions served a methodological objective (p. 485), but also an ethical objective related to the vulnerability of the population surveyed (Depoilly and Kakpo 2019, p. 224). The processing of the semi-open questions required a recoding corresponding to an a posteriori standardization (Depoilly and Kakpo 2019; Gauthier and Bourgeois 2016), which was carried out in the following way: each category was assigned a colour-coded numerical value (Paillé 1994); each category was then defined and put in relationship to the set of categories in order to highlight the possible links between them, and most importantly, in order to define a line of inquiry around each separate category. Quantitative and qualitative data was analyzed along the lines of each of these research areas (Gauthier and Bourgeois 2016). The first online questionnaire targeted funeral home directors and 51 responses were obtained. The second questionnaire was intended for employees of these funeral businesses: 89 people participated. Finally, 103 bereaved families responded to the third questionnaire, and 29 of them agreed to do a semi-structured telephone interview with the principal investigator of the study based on the fourth questionnaire. With respect to the online questionnaires, it is important to note that we did not focus on one population over another. The online questionnaires were made available to all employees and directors of the Corporation of Thanatologists, and to all bereaved families. We indicated that we were ready to assist anyone who was having difficulty completing any of the digital questionnaires. Our assistance was requested only twice. In addition, the website used to access the various questionnaires was entertaining and user-friendly. Also worth mentioning is the fact that the funeral industry

has developed a digital culture as a result of which its communication with bereaved families is increasingly being done through this medium. This context facilitated our study.

As regards the population of bereaved families, 81% of respondents were female, 27% were in the 51 to 70 age group, and 58% were rural Quebec residents. Seventy-five percent of respondents self-identified as Catholic and stated that 80% of their deceased also belonged to this religious tradition. These percentages explain in part the fact that 67% of the families requested a traditional, delayed funeral and demonstrate Quebec bereaved families' cultural and even religious attachment to tradition. A number of research areas emerged from this analysis. More specifically, the high percentage (67%) of bereaved families who requested a traditional delayed funeral paints a fairly uniform socio-cultural portrait of Quebec, including both rural and urban areas. At significant moments in their lives, in this case a loss, the families (in our context bereaved) seek a traditional religious environment and therefore a religious ritual to be part of these events. This observation has inspired us to conduct a new study, currently underway, on the type of ritual that bereaved families prefer in contemporary multicultural Quebec. Back to the point at hand, they are presented in the Table 3 below.

**Table 3.** Research areas.

| | Questionnaire 3 (Q. 3) Bereaved Families | | Telephone Interview (Q. 4) Bereaved Families |
|---|---|---|---|
| Area 1 | Helplessness | Area 1 | Helplessness |
| Area 2 | Delayed funeral | Area 2 | Delayed funeral |
| Area 3 | Space where one can be heard | Area 3 | Delayed grief |
| Area 4 | Need to grieve | Area 4 | Social group |
| Area 5 | Funeral rituals | Area 5 | Funeral Rituals |

The first research area reveals the sense of helplessness that families experience when forced to delay funerals. The second refers to the high percentage of families who choose to delay funerals. The third area of research highlights the fact that most families suffer due to delayed grief. The fourth area looks at both the need to grieve and the process of grieving in relation to the presence of a social group. There is no mourning without a community. The fifth line of research refers to an openness to rituals other than the traditional funeral; openness despite the drastic sanitary measures and rarely in lieu of a delayed funeral.

*4.2. Possible Solutions*

The five lines of inquiry encapsulate the research problem that is at the heart of the project commissioned by the Corporation of Thanatologists: a large majority of bereaved families experience delayed grief but, paradoxically, they do not prefer funeral rituals, certainly not in lieu of a funeral, referred to as a delayed funeral. Most of them therefore suffer due to delayed grief. Parallel to this, however, the analysis of the questionnaires shows a clear generational divide as regards delayed funerals. Indeed, the generation under 50 years of age favours funeral rituals and is very open to the proposed bimodal ritual. The same generational gap applies to questionnaires I and II targeting funeral home directors (Q. 1) and employees (Q. 2). Thus, a true generational divide emerges around the promotion of bimodal rituals. Insofar as they are considered in this study as a key solution and as a relevant pointer for a potential shift from religious to digital space during the COVID 19 pandemic, there are two types of shifts possible, both justified by two generations and/or distinct cultures. The first remains anchored in a religious posture that can be qualified as traditional. The second shift involves greater openness and a posture that integrates the digital world into the religious sphere. These distinct postures can be observed both among bereaved families and among employees and managers within the Corporation of Thanatologists. In light of these results, the third and last part of this article will focus on the type of shift that each posture induces.

## 5. Religion's Evolving Relationship to Cyberspace

For the purpose of analyzing the type of shift at work between the world of religion and that of the Internet, let's consider again the bimodal ritual as a pointer to a new relationship between religion and the Internet from an anthropological perspective in order to specify its structure, a structure shared by all rituals. Des Aulniers' (2007) analysis is of particular interest in this regard, because it focuses more on the structure and symbolic weight of ritual than on the mechanical and repetitive aspect thereof. Adopting the approach of social anthropologist (Thomas 1991), she defines ritual as "a range of acts and material signs with a highly symbolic content marking the experience of an event and a transformation that is perceived as mysterious and that calls for transcendence" (Des Aulniers 2007, p. 23). Let us see now how this structure can be applied to a bimodal, i.e., partly virtual ritual.

### 5.1. Virtual Performativity

When referring to bimodal ritual, what we have in mind is a ritual that respects the structure described by Des Aulniers (2007) and that is performed both in person and remotely, in other words, in the digital world. In her definition of ritual, Des Aulniers associates acts, material signs and their "highly" symbolic content with a view to overcoming a situation, delayed grief, for example. This structure therefore applies to the entirety of the bimodal ritual. But why would the digital mode of ritual have a particular performativity?

Acts performed virtually are free from the territorial and temporal barriers (Douyère 2015) that characterize sacrament and traditional liturgical gesture. There are no borders stopping the bereaved person living in Dubai, in the United Arab Emirates, from performing this gesture at a funeral ritual held in person in Montreal for his or her deceased loved one. Moreover, in digital space, it is doubly effective because it can be performed in real time. In addition, although with Des Aulniers we've left aside the mechanical aspect of the ritual, digital technology nonetheless allows the gesture to be repeated an infinite or almost infinite number of times. This gesture can be virtually placing one's hand on an equally virtual urn. Or virtually laying a wreath of flowers by the urn which, at that moment, is physically displayed at the funeral home for those who wish and are able to be there. The same is true for material "signs" that the bereaved in Dubai can make for their deceased loved one whose remains are in Montreal. This sign can be a hug—a virtual hug—but also an image or a text following Douyère's (2015) analysis, and it too is unrestricted by time or space and endlessly repeatable. Moreover, the performativity of this digital modality of ritual is multiplied tenfold insofar as the "disintermediation" leading to the abolition of all forms of authority that Douyère applies to DICT is also true in relation to ritual.

When implemented as a solution for achieving closure, the performativity of digital ritual is all the more effective. As argued by Howard (2011), the digital world is by definition a world of representation, i.e., of the image; it relates to the sphere of interpretation and the mechanics of the interpretation made by the bereaved in finding closure is crucial. Throughout the stages of the grieving process, grieving individuals can go back to the digital ritual as needed. This egological notion of digital ritual was emphasized by Douyère (2015) in relation to "undifferentiated believers." The closure gained on one's own from representation to interpretation is achieved at different levels according to. The pedagogical level refers to the accessibility of the digital ritual (whenever the bereaved person wishes). The meditative level is also accessible in cyberspace when the person feels ready to go through the various stages of grief. The mystagogical dimension depends on the bereaved person's spiritual and religious, or secular, code of reference. The process is the same for the semiotic dimension (Howard 2011). Why then should we associate the digital and the physical?

The purpose of the face-to-face aspect of the bimodal ritual is to reach out to a significant part of the population that will never be totally satisfied with a purely pictorial community (Howard 2011). The traditional religious tradition appears as a bulwark against

the culture of "religion online" (Helland 2005). In addition, the fourth line of research identified raises the question of the realism of ritual experienced in person. The COVID-19 pandemic has made us aware of this: community, i.e., family and friends, facilitates the stages of grieving and closure. For in order to mourn, it is important to be supported emotionally, to receive the needed embrace and appropriate expressions of sympathy. In this respect, it seems difficult to replace physical presence altogether, COVID-19 sanitary measures permitting.

It is worth noting that one of the advantages of the bimodal ritual is that its digital aspect preserves all the openness of the symbolic dimension–guardian of the relationship to transcendence–while its in-person aspect ensures the realistic anchoring of symbolism and hence the semiotic dimension inherent to all ritual.

### 5.2. What Type of Shift?

Following Campbell (2005), four paradigms systematizing the connection between religion and the digital world have been identified. The analysis put forward in this article allowed to specify a religious posture in relation to the digital world for each paradigm (see Table 1): Spiritual network induces an external posture in relation to the digital world; Worship space, an interior religious posture; Missionary tool, a downstream posture; and Religious identity, an upstream posture. If in line with Campbell (2005) and Douyère (2015), we consider religion as both an organized system of beliefs and a space of interaction par excellence, then bimodal ritual is a religious expression. It follows from there that the exercise of this bimodal ritual places religion at the centre of the digital world, marking the latter as a religious space, one that is highly performative and effective.

The complexity and richness of this ritual and hence of this new relationship between the traditional religious world and the digital religious world are owed to its bimodal character. For this type of ritual and for digital religion broadly speaking, the centre of inertia lies in the digital whereas the in-person aspect is considered peripheral. The bimodal form of ritual operates a two-fold shift from the religious to the digital: toward the digital by virtue of one modality and toward the peripheries by dint of the other modality. The shift between two worlds resulting from the pandemic as manifested through the bimodal ritual shows a hardening of the relationship between the traditional religious world and the open religious world. This cultural reality comes through quite clearly in the research conducted for the Corporation of Thanatologists. Consequently, the solution proposed, i.e., bimodal ritual, is in line with this cultural and metaphysical tension. The challenge here is to combine modalities with a view to qualifying digital religion (by dint of the ritual) without leaving the peripheries–institutional, traditional religious, cultural and societal that lie outside this digital religious revolution–to their own devices. The COVID-19 pandemic has expressed and exacerbated a cultural tension that had already manifested. It is not so much a question of a paradigm shift as it is a matter of merging together the first two paradigms identified by Campbell, spiritual network and worship space, in a religious posture that is metaphysically anchored in the virtual while soliciting the periphery. This is indeed what the typology of DICT use (Table 2) developed on the basis of Douyère's (2015) analysis highlights. Digital information and communication technologies (DICT) address both the social group, its institutions, the world of knowledge, and so on. Digital religion is not only effective, but also invasive. Therefore, this intrusion can be viewed as an exaggerated posture or as an opportunity for dialogue between digital religion and its periphery.

If the COVID-19 pandemic provokes a crystallization of the rift between traditional and digital religion, and more broadly between social culture and digital culture, it is above all indicative of a pre-existing cultural tension. There are two possible hypotheses for the post-pandemic era. The first is the acceptance of this new relationship to the digital world by traditional religion which is on the periphery. The bimodal ritual proposed by way of a solution would in this case be the precursor of a culture to come. However, it is also possible to assume that digital religion and its weapon of effectiveness will conquer and

own peripheries of all kinds: institutional, economic, cultural and sanitary. Whereas the first hypothesis proceeds from the association of paradigms 1 and 2 proposed by Campbell, the second is a consolidation of the Worship space and begs the question of whether religion can survive such a culture of effectiveness?

## 6. Conclusions

The question of the effectiveness of religion connects in a cross-cutting fashion to the purpose of this article. The latter focused on the possible shift in the relationship between traditional religion and the digital continent during the COVID-19 pandemic, and analysis of the bimodal ritual confirms that everything in the digital realm, its structure, DICT, rituals, everything refers to a language, a posture and a culture of effectiveness. What does this tell us about digital religion? Above all, what does this type of effectiveness induce in terms of dealing with human crisis and even spiritual crisis? To provide an example, and by way of conclusion, let's apply the effectiveness linked to digital ritual to the problem of delayed grief. If the digital world is increasingly interested in grief and closure–in relation to which the work of an interdisciplinary research network (Christensen and Gotved 2015) and the group *Death Online Research* (deatonlineresearch.net, access on 12 April 2021) deserve to be pointed out–the question of closure within digital religion and its rituals is a topic that warrants further research. What is referred to as complicated grief (Ben-Cheikh et al. 2020), atypical grief (Bacqué and Hanus 2016), anticipatory grief (Philippe and Touren-Hamonet 2021) or grief before death (Fasse et al. 2013), all differ from delayed grief in that the process has already begun. Delayed grief, on the other hand, is by definition suspended grief associated with avoidance (Stroebe et al. 2017) and stunning. Therapists, counsellors and even religious persons find themselves at a loss when faced with this type of grief. How do we question without provoking? This is where digital religion and bimodal ritual are uniquely effective. Since interpretation in the virtual community is personal (Helland 2005), it allows each of its members to choose not only the moment of interpretation–in this case the death of a loved one—but also its meaning. Digital religion invites through images and ritual and the subject decides. Douyère (2015) warns against this "ego-localization." Free of mediation and therefore authority, digital religion is effective because the initiator is the subject himself or herself. In other words, whereas traditional religion imposes sacraments and discourses from the outside, digital religion enables a self-centred hermeneutics. This is where the basis of its effectiveness lies. From a long-term perspective, however, in relationship to grief for example, it is possible to discern the flaws of this type of effectiveness. In the stages of grief taught by Monbourquette and d'Aspremont (2016) and Monbourquette (2004), effectiveness lies in the linearity of counselling. According to Stroebe's grieving model (Stroebe et al. 2017), the oscillation that takes place requires the physical presence of the therapist. In a nutshell, this analysis of the shift between traditional and digital religion raises a crucial question: Unbeknownst to itself, does digital religion introduce an effective but brittle relationship to reality?

**Funding:** This research received no external funding.

**Institutional Review Board Statement:** Not applicable.

**Informed Consent Statement:** Not applicable.

**Conflicts of Interest:** The authors declare no conflict of interest.

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
