# Peer review of "Study of the Changing Relationship between Religion and the Digital Continent—In the Context of a COVID-19 Pandemic"

_religions, doi:10.3390/rel12090736_

Round 1
Reviewer 1 Report
This is an interesting article on the emerging landscape of digital religion. Particularly in relation to the COVID-19 pandemic, the article will provide a useful reference point for further research on the shifting landscape of religious practice and the ways in which our increasingly digital world influences how humans practice religion and how effective and meaningful this practice will be in the lives of those who take part in these digital rituals.
Author Response
Here is the document with the adjusted bibliography. thank you

Reviewer 2 Report
Thank your for this interesting and challenging article on delayed grief. Your text is an important contribution to the discours on 'religion' in times of 'online encounters' - encounters with the other and with The Other.
The main question addressed is the relation between online mourning and ‘real life’ mourning. In my view an extreme interesting and important question in times of Covid-19, and in particular in future times for us as cosmopolitan citizens of a global world. This article gives food for thought regarding the process of mourning and the importance of people to share feelings with - people in the near life world ‘next door’ as well as people farther away in the virtual world. New, and adding to what we know already of this subject, is that online funerals can be reviewed and reviewed, and in that way can be helpful for a ‘healthy’ mourning process. Regarding the way this article is written: it shows the ‘hand of the master’ - a scholar who is familiar and who has internalised the criteria of a logic structure, that means the conclusion and the discussion are in line with the findings.Author Response
Here is the document with the adjusted bibliography. thank you

Reviewer 3 Report
This text reveals relevants features of the relation between religion and the virtual world. It depicts the new ways of organisation the ritual in context of Covid19.
Mi doubts:
- To introduce in the bibliography many references used in the text, that they don't appear in it.
- To extend the reflection in order to clarify if the new possibilities oponed by the virtual world has influence on the religion. What happens with the sacred?, and with the relationship between man and the mistery?
- To rethink the outcomes of a ritual that it depends totally of the individual will.
Author Response

(The authors gave the same response as above.)
